# Canine Distemper Virus in Sardinia, Italy: Detection and Phylogenetic Analysis in Foxes ^[note 1]^

**DOI:** 10.3390/ani14213134

**Published:** 2024-10-31

**Authors:** Elisabetta Coradduzza, Fiori Mariangela Stefania, Davide Pintus, Luca Ferretti, Alice Ledda, Gian Simone Chessa, Angela Maria Rocchigiani, Giada Lostia, Renata Rossi, Maria Giovanna Cancedda, Simona Macciocu, Marcella Cherchi, Daniele Denurra, Antonio Pintore, Roberto Bechere, Flavia Pudda, Marco Muzzeddu, Maria Antonietta Dettori, Angelo Ruiu, Paolo Briguglio, Ciriaco Ligios, Giantonella Puggioni

**Affiliations:** 1Istituto Zooprofilattico Sperimentale della Sardegna, 07100 Sassari, Italy; mariangela.fiori@izs-sardegna.it (F.M.S.); davide.pintus@izs-sardegna.it (D.P.); giansimone.chessa@izs-sardegna.it (G.S.C.); angelamaria.rocchigiani@izs-sardegna.it (A.M.R.); giada.lostia@izs-sardegna.it (G.L.); rossirena73@gmail.com (R.R.); mariagiovanna.cancedda@izs-sardegna.it (M.G.C.); simona.macciocu@izs-sardegna.it (S.M.); marcella.cherchi@izs-sardegna.it (M.C.); danidenur@virgilio.it (D.D.); antonio.pintore@izs-sardegna.it (A.P.); roberto.bechere@izs-sardegna.it (R.B.); mariaantonietta.dettori@izs-sardegna.it (M.A.D.); angelo.ruiu@izs-sardegna.it (A.R.); ciriaco.ligios@izs-sardegna.it (C.L.); giantonella.puggioni@izs-sardegna.it (G.P.); 2Pandemic Sciences Institute, Big Data Institute, Li Ka Shing Centre for Health Information and Discovery, Nuffield Department for Medicine, University of Oxford, Oxford OX1 2JD, UK; luca.ferretti@gmail.com; 3UK Health Security Agency, Colindale, London NW9 5EQ, UK; alice.ledda@gmail.com; 4Centro di Recupero della Fauna Selvatica Ferita (C.A.R.F.S), Bonassai, Agenzia Forestas Sardegna, 07100 Olmedo, Italy; fpudda@enteforestesardegna.it (F.P.); muzzeddum@tiscali.it (M.M.); 5Clinica Veterinaria Duemari, 09170 Oristano, Italy; vetoristano@gmail.com

**Keywords:** canine morbillivirus, *Vulpes vulpes ichnusae*, lineage

## Abstract

Canine distemper is a contagious infectious disease caused by the canine distemper virus (CDV). The disease affects carnivores and is present in different geographical areas. This work aimed to characterize CDV strains infecting foxes circulating in Sardinia (Italy) from 2014 to 2023, using a multidisciplinary approach. Our sequences, similar to those of central Italy, form a group very close to the European group of wildlife sequences, which leads us to propose it as exclusive to Sardinia (Sardinia-Wildlife). This study enriches knowledge of the strains responsible for natural CDV infection in this species and of the surveillance of their spread in the territory itself. It is important to continue monitoring the distribution of the virus and typing the strains to better understand the dynamics of virus circulation among wild and domestic carnivores.

## 1. Introduction

Canine distemper virus (CDV) is the etiological agent of a highly prevalent viral infectious disease of carnivores, which could seriously lead to a threat to the conservation of the affected species worldwide [1,2]. The virus affects domestic and wild animals, particularly carnivores [2], threatening some endangered species [3,4]. CDV outbreaks have occurred over the past few years in canids, mustelids, felids, ursids, and even non-human primates [5,6,7,8,9,10,11,12].

This disease, known in dogs as early as 1760, is classified as highly contagious and febrile [13]. CDV has a high tropism for epithelial, lymphoid, and nervous system cells, causing a systemic infection that involves the respiratory, digestive, urinary, lymphatic, skin, skeletal, and central nervous system [14]. This virus, isolated in 1905 by Henri Carrè [15], is a member of the genus *Morbillivirus* [16] with a non-segmented single-negative-stranded RNA (ssRNA) [14]. The genome is about 16 KB in length and its structure includes six transcription units (ORFs) encoding for six structural proteins (two glycoproteins: hemagglutinin (H) and fusion (F) proteins; one envelope-associated matrix (M) protein; two transcriptase-associated proteins: phosphoprotein (P) and large polymerase (L) protein; and one nucleocapsid (N) protein [17,18]), separated by intergenic untranslated regions (UTRs) [16]. In particular, hemagglutinin (H) is an integral membrane glycoprotein that facilitates the virus binding to the host cell membrane and, compared with other morbilliviruses, it has been shown to have higher heterogeneity [19] permitting clustering viral strains of CDV into at least nine lineages according to their geographical distribution: America 1 (NA1), America 2 (NA2), Europe 1/South America 1 (EU1/SA1), Europe 2/Europe-Wildlife (EU2), Europe 3/Arctic-like (EU3), Asia 1 (AS1), Asia 2 (AS2), South Africa (ZA), and South America 2 (SA2) [16,18,20]. In Italy, EU1/SA1 (found in both domestic dogs and wild carnivores) is particularly prevalent in the north [19,21,22,23], while in the center and south, the most representative lineages are EU2 (rarely found in domestic dogs) and EU3 (found mainly in Italian dogs) [5,24,25]. Within the Europe/South America-1 lineage, two distinct subclades have been identified in northern Italy in recent years [5,19,26]: clade a, which originates from ancient epidemic waves in both domestic and wild carnivores, and clade b, which consists of CDV strains most likely imported from the Balkan region.

The Sardinian fox (*Vulpes vulpes ichnusae*) is endemic in Sardinia and is anthropized; it is found in forest and scrubland environments, agro-systems, and human settlements [27]. Several phylogeny studies relate the viral sequences of distemper strains from domestic dogs and wild carnivores, suggesting horizontal transmission between species through aerosols and exudates [24,25,26,27,28,29,30]. CDV does not always appear to have a particular impact on foxes [19]: long-term persistence of CDV can occur for decades, sometimes without fatal consequences [8,31] because animals with robust immune responses can recover by developing lifelong immunity to reinfection. However, they play an important role as reservoirs for highly susceptible species. Early detection in wildlife species is therefore particularly important to allow the typing of circulating CDV strains.

This work aimed to characterize phylogenetic distemper strains infecting foxes in Sardinia, Italy, from 2014 to 2023, i.e., [1,12,19]. With this objective, we sequenced the H gene of all CDV virus strains isolated from infected foxes, i.e., [26,32]. In addition to the phylogenetic analyses, histological and immunohistochemical examinations were performed in foxes to better characterize the pathological features of the viral strains affecting this species.

## 2. Materials and Methods

### 2.1. Animals and Sampling

A total of 42 animals were examined: 41 foxes (Figure 1) and 1 dog. The addition of the canine sequence (sample already analyzed in our laboratory during routine analysis) aimed to understand possible connections from a phylogenetic point of view with fox sequences. All animals underwent necropsy examination and, for each one, either the brain, lung, spleen, or eyelid were sampled. Then, the most representative sample from a molecular point of view was selected for sequencing (Table 1). Where there were no limitations such as post-mortem alterations or progressed decomposition of the cadavers, the brain and the lung were collected for histological and histochemical exams. 

### 2.2. Pathology and Immunohistochemistry

Collected organs were fixed in 10% formalin and embedded in paraffin for histological evaluation. According to routine laboratory protocols, 4 μm thick sections were stained using the ST Infinity Haematoxylin & Eosin Staining System (Leica Biosystems, Richmond, IL, USA) and then examined using a light microscope (from 50× to 400× of magnification). Furthermore, on selected brain and lung tissue sections, immunohistochemistry (IHC) was performed using a CDV-specific primary antibody (Bio-Rad monoclonal mouse anti-canine distemper virus antibody, clone DV2-12, MCA 1893, Hercules, CA, USA). Moreover, IHC was performed on selected brain and lung sections for different clusters of differentiation markers (CD immunophenotyping) of inflammatory cells. Primary antibodies were rabbit monoclonal CD3 (Clone 7.2.38, DAKO Agilent, Santa Clara, CA, USA), mouse monoclonal CDacy79 (Clone HM57, DAKO Agilent, Santa Clara, CA, USA), monoclonal mouse CD163 (Clone EDHu-1, Bi-Rad, Hercules, CA, USA). Working solutions were 1:1500 for CD3, 1:750 for CDacy79, and 1:800 for CD163.

Sections were deparaffinized and rehydrated through several passages using alcohol and then thoroughly rinsed in water. Heat-induced antigen retrieval was performed in target citrate buffer pH 6.0 (Thermo Fisher Scientific, Waltham, MA, USA) or in target retrieval solution pH 9.0 (Dako Agilent, Santa Clara, CA, USA) to expose antigen epitopes. To quench endogenous peroxidase activity, the slides were immersed in 0.3% H_2_O_2_ for 35 min at room temperature and then incubated overnight at 4 °C using the primary antibodies as previously described. Finally, immunoreactivity was detected via the biotin–avidin–peroxidase method using 3,3′-diaminobenzidine as a chromogen. Later, slides were counterstained with Mayer’s hematoxylin and mounted by cover slides. In each IHC run, adequate positive and negative control tissues were included. Moreover, brain, lung, conjunctival mucus, and blood in EDTA were subjected to biomolecular investigation to diagnose the presence of CDV by RT-PCR amplification. The affected foxes that showed gastrointestinal signs were also subjected to molecular investigation using PCR for Parvovirus [33], Nested RTPCR for Coronavirus [34], and Real-Time PCR for pathogenic Leptospira ssp. [35].

### 2.3. Nucleic Acid Extraction, RT-PCR Amplification, and Sequencing

First, a highly conserved portion of the gene that encodes for the nucleoprotein (N gene, of 287 bp) was chosen to perform a screening RT-PCR suitable for detecting all CDV strains [36]. Subsequently, we amplified the H gene, using three PCRs to cover the entire sequence (sequenced samples in Table 1 and primer set in Table 2). The amplification covered areas from the F gene’s end part to the L gene’s beginning part, which flanks the H gene in the sequence.

RT-PCR was performed using the Superscript One-Step RT-PCR Kit with Platinum Taq (Thermo Fisher Scientific, Waltham, MA, USA), according to the manufacturer’s directions.

The PCR products were purified using the CleanSweep PCR Purification kit (Thermo Fisher Scientific) and sequenced for both forward and reverse strands (using the same primers used for PCR) using a Sanger Sequencing 3500 Series Genetic Analyzers Terminator 3.1 apparatus (Applied Biosystems, Waltham, MA, USA).

### 2.4. Sequence and Phylogenetic Analyses

Sequences were aligned using BioEdit 7.2.5 [37] and MEGA 7.0 [38] softwares. A maximum likelihood optimized search was employed for the phylogenetic reconstruction of Sardinian and international sequences using IQ-TREE version 1.6.12 [39] with a GTR + Gamma evolutionary model with standard parameters and ultrafast bootstrap approximation. In addition, Bayesian reconstruction of the dated phylogeny of Sardinian sequences was performed using BEAST version 1.10.4 [40] with a Skygrid relaxed-clock model with annual resolution 2003–2023 and 100 million steps for the MCMC chain. Molecular distances for nucleotide and amino acid sequences were computed using R version 4.3.2 and the ape package version 5.7 with default parameters.

## 3. Results

### 3.1. Clinic, Pathology, and Immunohistochemistry

Sick animals showed sensory depression and ataxia, as well as blindness or severe vision alteration in both eyes, and some of them had hematochezia. All foxes died within 2 to 48 h in the recovery center. Gross examination revealed severe chronic wasting, muco-purulent conjunctivitis, and nasal discharge (Figure 2). Congestion and/or hemorrhage in the lung sometimes associated with sero-hemorrhagic fluid in the pericardium and mild congestion in the vessels of the cerebral cortex were observed.

Histologically, the brain lesions were characterized by moderate to severe demyelination, and the diffuse mild presence of mononuclear inflammatory cells involving the meninges. In-depth examination revealed focal to multifocal vacuolation (demyelination) of the white matter and an increased number of reactive astrocytes, macrophages, and microglial cells, sometimes organized in nodular patterns. A marked spongy state with small or large vacuoles was observed in the demyelinated lesion areas (Figure 3a,b). Occasionally, necrotic neurons with shrunken and eosinophilic cytoplasm and lymphoid cells around blood vessels were found. No viral nuclear inclusion bodies were observed. Lesions were observed in the gray and white matter of serial brain sections but were more severe in the cerebellum (Figure 3c). In the cerebellar sections, the loss of Purkinje cells was segmentally lost, and the progressive loss of myelin characterized by vacuolization of the white matter was evident. The mononuclear infiltrate cells observed in the meninges (Figure 3d) were immunohistochemically characterized mainly by CD163+ macrophages and CD3+ T lymphocytes (Figure 4), whereas CD79+ cells were few or absent.

Neurons and a limited number of glial cells from the cerebral cortex showed strong and specific CDV immunolabeling. Moreover, CDV-positive staining was associated with the endothelial cells of numerous vessels in the cerebral parenchyma and meninges. Confirming the severity of lesions determined in the brain sections, high immunoreactivity for CDV was revealed in the cerebellum section, associated with the molecular and granular layer, and in the body and axons of the Purkinje cells (Figure 5). Histologically, in the lung, pulmonary lesions were characterized by diffuse and severe interstitial pneumonia. As confirmed by CD immunophenotyping, an infiltrate of macrophages (CD163+) and lesser numbers of T lymphocytes (CD3+) associated with edema and erythrocytes enlarged the alveolar septa. Very few neutrophils were also observed. Blood vessels were observed in some pulmonary sections occluded by inflammatory cells and debris. Syncytial giant cells were also observed. Finally, hypertrophy of smooth muscle cells around the airways and blood was determined in some of the examined lungs (Figure 6a,b).

Immunohistochemical staining of the lung highlighted the presence of the CDV antigen in macrophages and lymphocytes, as well as in the epithelial cells of the bronchi and bronchioles (Figure 6c,d).

### 3.2. Nucleic Acid Extraction, PCR Amplification

In all subjects, it was possible to detect the presence of viral RNA by RT-PCR of the N gene. No other etiological agents were revealed in the samples. The positivity was confirmed by direct sequencing of the amplified H gene.

### 3.3. Sequencing and Phylogenetic Analyses

As can be seen from the reconstruction of the global CDV phylogeny (Figure 7), all our Sardinian sequences from foxes belong to a single clade. Only one other sequence from the dog sample, sequenced in this work, is in the same clade, specifically in the middle, suggesting a transmission involving a host jump from foxes to dogs [31,41].

As shown in Figure 8, the Bayesian reconstruction of the time-calibrated Sardinian phylogeny points to a more recent common ancestor of all sampled Sardinian sequences circulating in 2008 (mean 2008.6, median 2009.3, 95% 95% HPD interval 2002.7–2012.9), suggesting that the lineage currently circulating in Sardinia was introduced to the island a few years before the time when the first sequenced samples were obtained, i.e., 2014 (Figure 9).

A Bayesian reconstruction of the effective population size (Figure 10) suggests that the circulation of CDV in the Sardinian epidemic may have gradually increased in the first decade since its introduction, possibly followed by a slight decrease in recent years.

The maximum likelihood phylogenetic reconstruction places this Sardinian clade close to the Europe-Wildlife lineage (Figure 7). However, it is significantly divergent from published sequences. The closest sister clade contains samples from different mammals from central Italy (foxes and badgers) [25,26], and the two clades are significantly divergent (5.0% at nucleotide level, 4.7% at amino acid level, Table 3). This suggests that these two clades represent two different lineages. Given the geographical proximity, it is plausible that the Sardinian clade may have originated from viruses from the continental part of Italy introduced into the island.

Together, the Sardinian and other Italian sequences represent a sister clade of the published Central European sequences classified as belonging to the Europe-Wildlife lineage. Note that these sister clades are very different from sequences belonging to the Europe-Wildlife lineage from Germany and Austria: the nucleotide divergence is 5.0% and 5.2% for Sardinian and Italian sequences, respectively, and the amino acid divergence is 5.5% and 5.3% (Table 3). This divergence is comparable to the divergence between different CDV lineages [20]. Most likely, Sardinian sequences belong to a novel “Sardinia-Wildlife” lineage.

## 4. Discussion

The phylogeny, based on the H gene, clusters the Italian strains into three lineages: Europe/South America-1, Europe-Wildlife, and Arctic-like [5,24,25,42]. In northern Italy, two distinct subclades within the Europe/South America-1 lineage have been identified in recent years [5,19,26]: clade a, which originates from ancient epidemic waves in both domestic and wild carnivores, and clade b, which consists of CDV strains most likely imported from the Balkan region. These data show the crossover between domestic and wild carnivores for strains capable of jumping species and the diverse behavior of virus strains, some of which remain specific to the original host. However, to date, the knowledge and understanding of how the virus spreads in southern Italy, both in domestic dogs and wildlife animals, is still very poor and limited [5,20,43,44,45]. Intending to contribute to the current knowledge on the spread of canine distemper virus in Italy, this study provides a molecular characterization of CDV strains based on the H gene sequence obtained from viral samples isolated from foxes in Sardinia and a comparison of them with currently available CDV sequences.

As previously reported, the central nervous system and lungs were two of the main CDV target organs in carnivores [14]. The RT-PCR associated with the immunohistochemical results revealed that, with a high probability, CDV was the etiological agent of the neurological signs in the enrolled foxes in this study. The pulmonary histological lesions and immunohistochemical detection of CDV in the mucosa of the bronchial tree support the potential viral excretion through nasal discharge.

The Sardinian sequences are similar among them but diverge by about 5% from other known CDV sequences, both at the nucleotide and amino acid levels. This divergence is similar to the divergence between different CDV lineages, such as Asia-4 and the recently discovered Asia-6 [46], suggesting a significant degree of isolation among viral populations in Sardinia and elsewhere (Italy and Central Europe). Given their geographical isolation and the exhaustive sampling of viruses circulating on the island, at least in the unique wild carnivore species present, these Sardinian sequences represent a novel “Sardinia-Wildlife” lineage circulating in Sardinia.

The sister clade of Sardinia-Wildlife contains only sequences sampled from wild mammals from central Italy [26], which appear to be representative of the sequences currently circulating in central Italy, given that they include many samples from the Romagna region on the northern side but also a similar sample from the Abruzzo region on the southern side of central Italy [47], about 300 km away from each other. These sequences have about 5% divergence at the nucleotide and amino acid levels from both Sardinia-Wildlife and other Europe-Wildlife sequences from Central Europe. Given their similarity and divergence compared to other CDV sequences, we suggest that central Italian sequences may represent another potential “Italy-Wildlife” lineage, even though the incompleteness and unrepresentativeness of the geographical sampling in central and northern Italy limits the validity of this claim. A more extensive and exhaustive sampling of intermediate geographical regions as a wider number of sequences is needed to properly define the phylogenetic relation between these Italian viruses and Europe-Wildlife.

To gain a better understanding of the evolutionary dynamics of CDV strains in foxes and dogs by studying their amino acid sequence, we evaluated in particular the receptor binding sites 530 and 549 of hemagglutinin, which McCarthy in 2007 [48] had already hypothesized to play a functional role in determining cell tropism since the spread of the virus in non-dog hosts is most associated with amino acid substitutions at these sites. At site 530, dog strains usually present amino acid G or E, whereas amino acid substitutions N, R, D, or S are associated with non-dog hosts. All our strains, including the dog, present the amino acid N at this site, leading to the hypothesis that fox and dog strains have undergone common evolution. This hypothesis would also be confirmed by the 549 binding site in the SLAM binding region, which presents the amino acid Y (typical of canine strains and found in all our sequences) in all of them.

## 5. Conclusions

This study shows the dynamics of CDV infection within the fox population in Sardinia, enriching our knowledge of the strains responsible for natural CDV infection in this species and of the surveillance of their spread in the territory. Since the number of positive cases in foxes is quite worrying, leading to high exposure to the virus in environments where other susceptible species circulate, it is important to continue monitoring the distribution of the virus and typing the strains to contain its spread and to better understand the dynamic of the virus’s circulation among wild and domestic carnivores.

## Figures and Tables

**Figure 1 animals-14-03134-f001:**
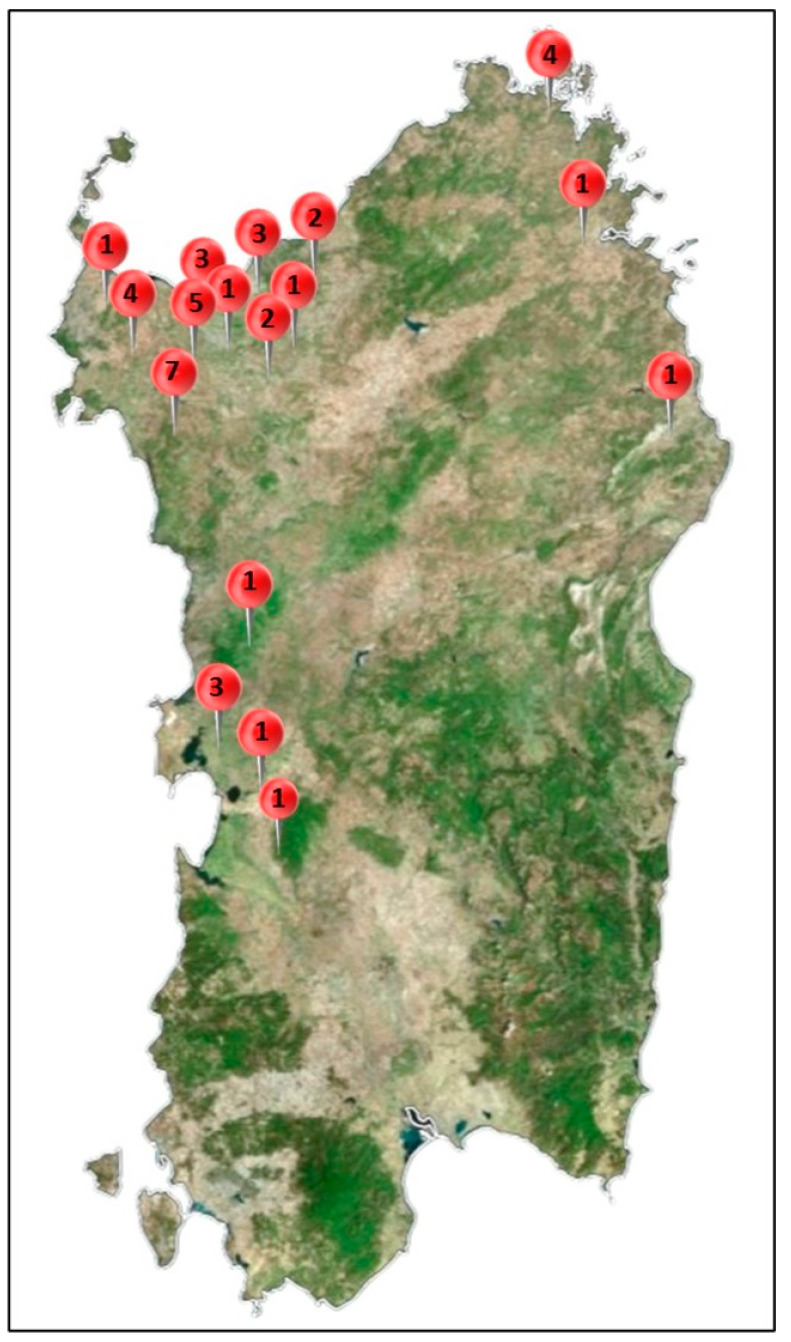
Map of the study area showing the locations where fox samples were collected. The numbers indicate how many of the 41 samples examined in this study were collected in that area.

**Figure 2 animals-14-03134-f002:**
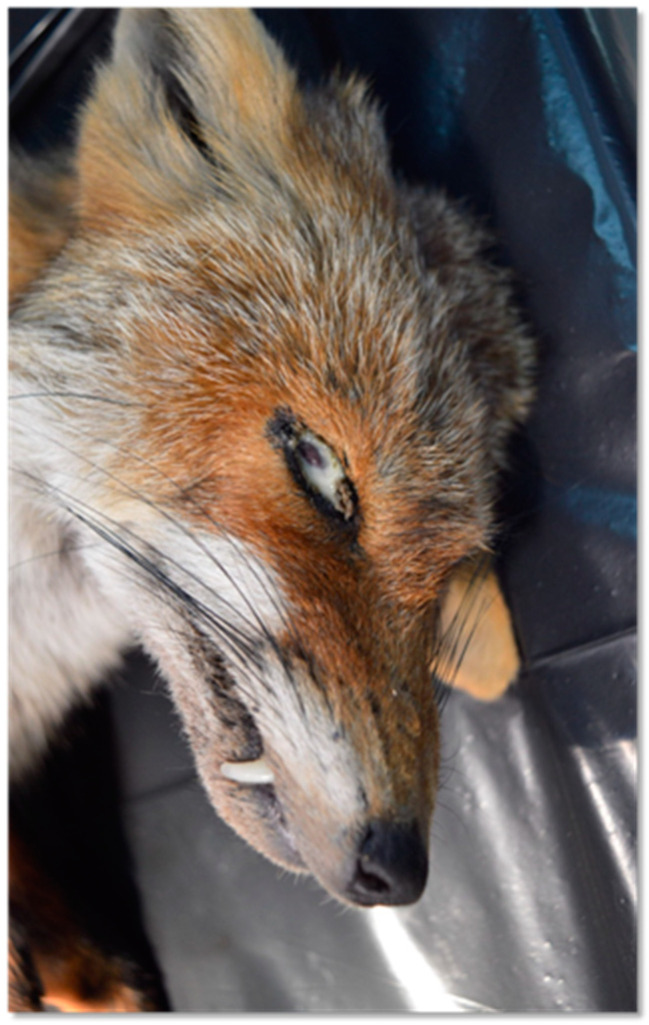
External examination of a fox with CVD. Eye with muco-purulent conjunctivitis.

**Figure 3 animals-14-03134-f003:**
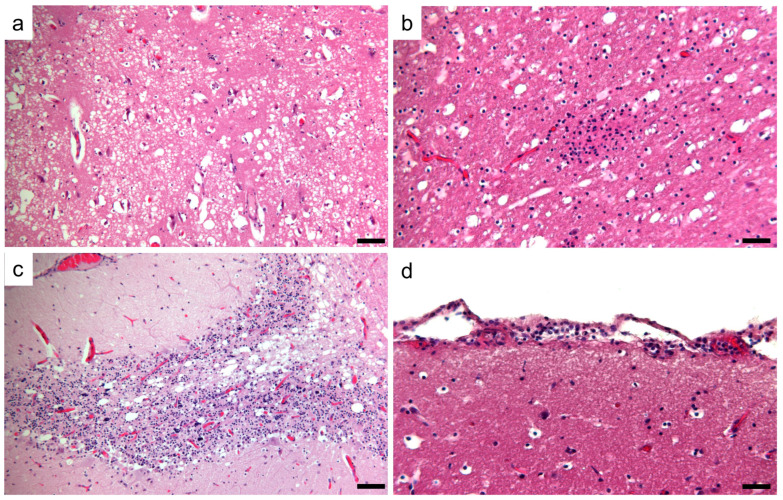
Photomicrographs of representative histological brain sections of CVD-infected foxes. (**a**) Cerebral cortex: severe demyelinating encephalitis with vacuolization of the white matter due to myelin sheath edema. (**b**) Cerebral cortex: nodular gliosis in a moderate demyelinating area of the cortex. (**c**) Cerebellum: demyelinating areas in the substantia alba of the cerebellum associated with the loss of Purkinje cells and the hypocellular granule cell layer. (**d**) Cerebral cortex: mild meningeal lymphohistiocytic infiltrates. Hematoxylin and eosin staining. Scale bar 100 µm.

**Figure 4 animals-14-03134-f004:**
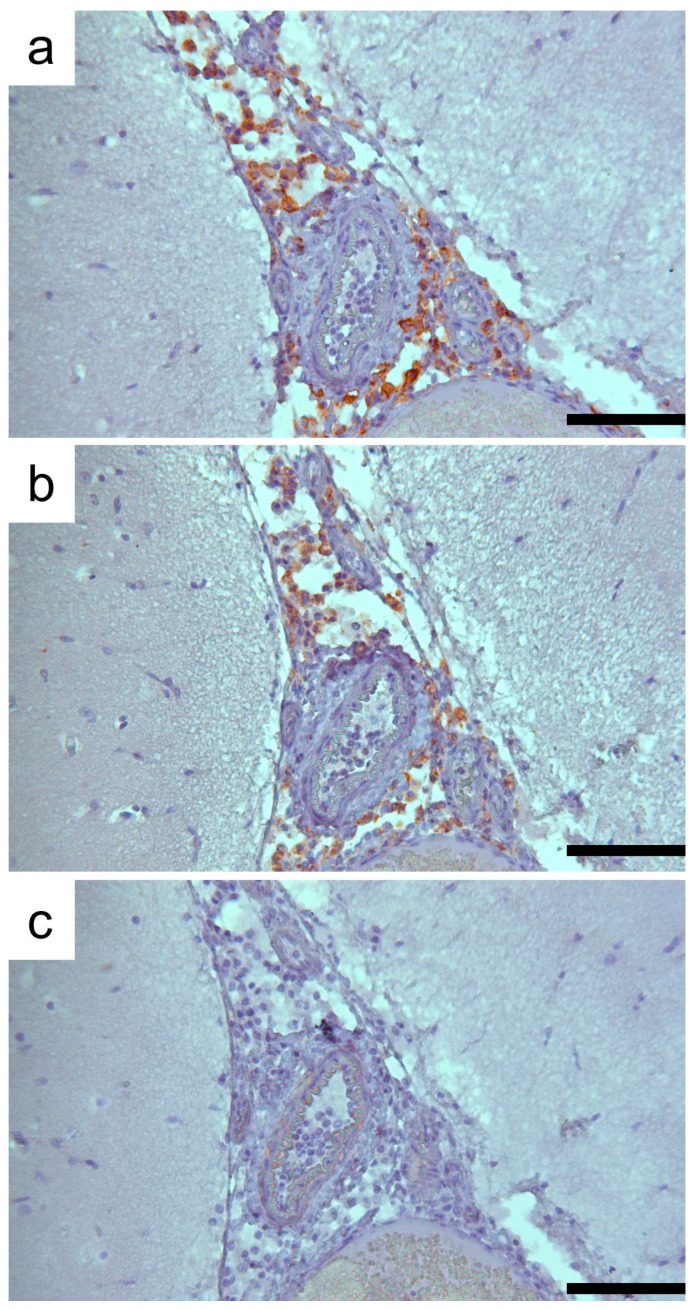
CD-immunophenotyping of the meningeal inflammatory cells. (**a**) Immunohistochemical labeling of CD163+ macrophages. (**b**) Immunohistochemical reactivity of CD3+ T lymphocytes. (**c**) No CD79+ B lymphocytes were revealed. 3-30-diaminobenzidine (DAB) chromogen with Mayer’s hematoxylin counterstain. Scale bar 100 µm.

**Figure 5 animals-14-03134-f005:**
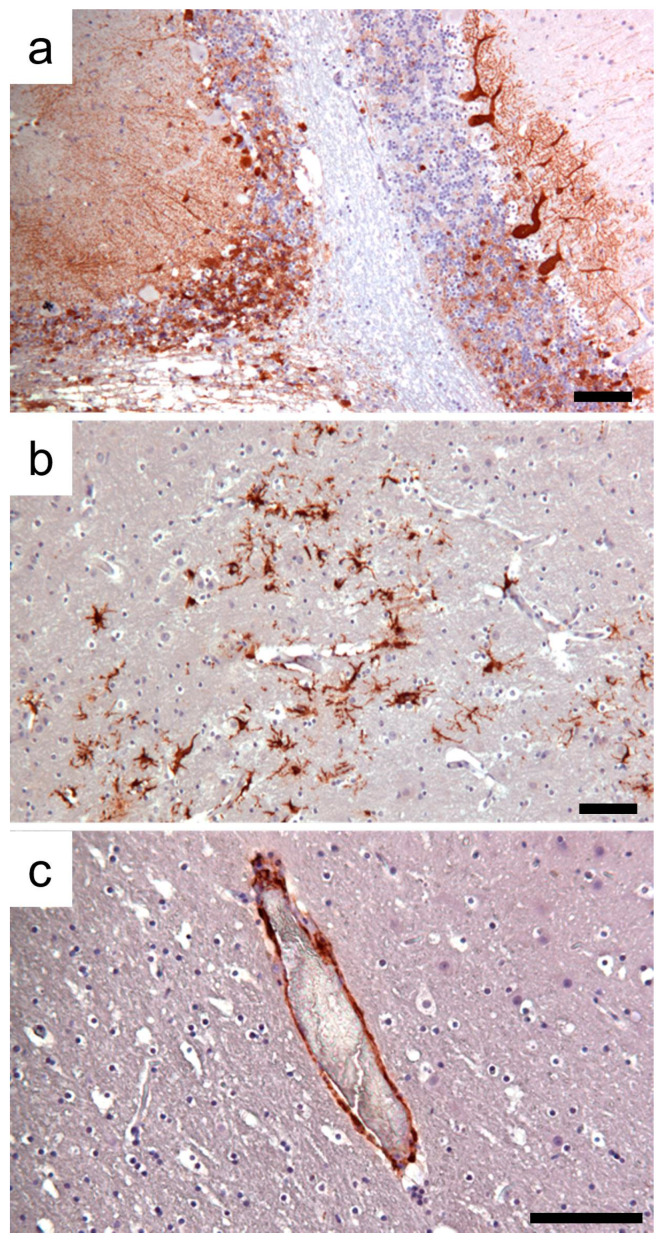
Photomicrographs of representative CDV immunoreactivity in brain sections. (**a**) Cerebellum: high immunoreactivity in the granular and molecular layers, and in the few visible Purkinje cells. (**b**) Intense and specific immunohistochemical CDV expression in neurons. (**c**) CVD immunohistochemical staining associated with endothelial cells. 3-30-diaminobenzidine (DAB) chromogen with Mayer’s hematoxylin counterstain. Scale bar 100 µm.

**Figure 6 animals-14-03134-f006:**
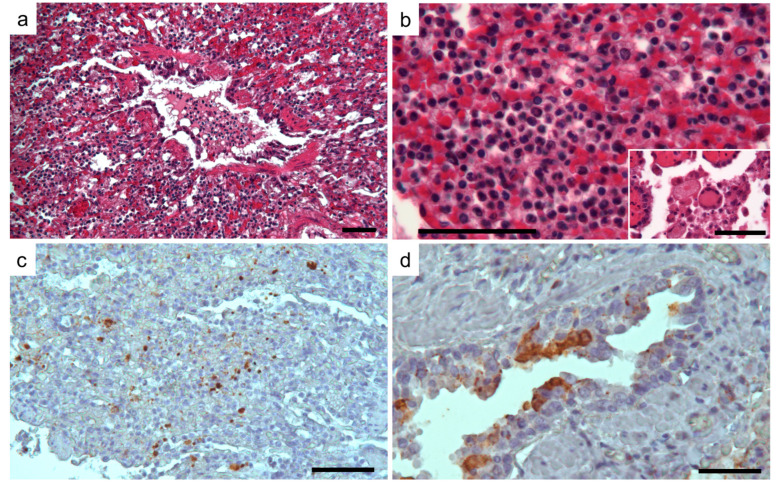
Photomicrographs of representative histological lung sections of CVD-infected foxes. (**a**,**b**) Two different magnifications of interstitial pneumonia are characterized by debris and inflammatory cells in the bronchiolar lumen and by thickening of the alveolar septa with macrophages, lymphocytes, and erythrocytes predominantly. The syncytial cell is revealed in the insert. Hematoxylin and eosin staining. Scale bar 100 µm. (**c**,**d**) Positive immunolabelling is associated with inflammatory cells, resembling macrophages, and in the bronchiolar epithelial cells. 3-30-diaminobenzidine (DAB) chromogen with Mayer’s hematoxylin counterstain.

**Figure 7 animals-14-03134-f007:**
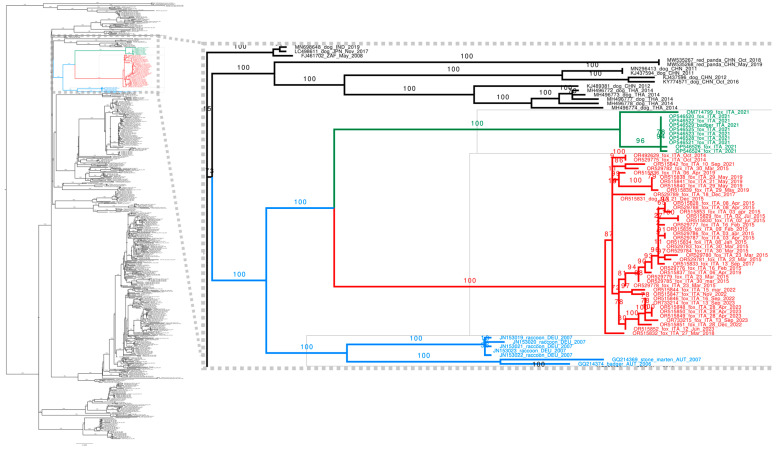
Maximum likelihood reconstruction of the global CDV phylogeny. Sardinian sequences sampled in this work are shown in red, while those belonging to the European wildlife lineage are in blue and green; the latter color used is for the Italian sequences belonging to the sister clade to the Sardinian one. Branches are annotated by ultrafast bootstrap values (splits are considered reliable if >=95).

**Figure 8 animals-14-03134-f008:**
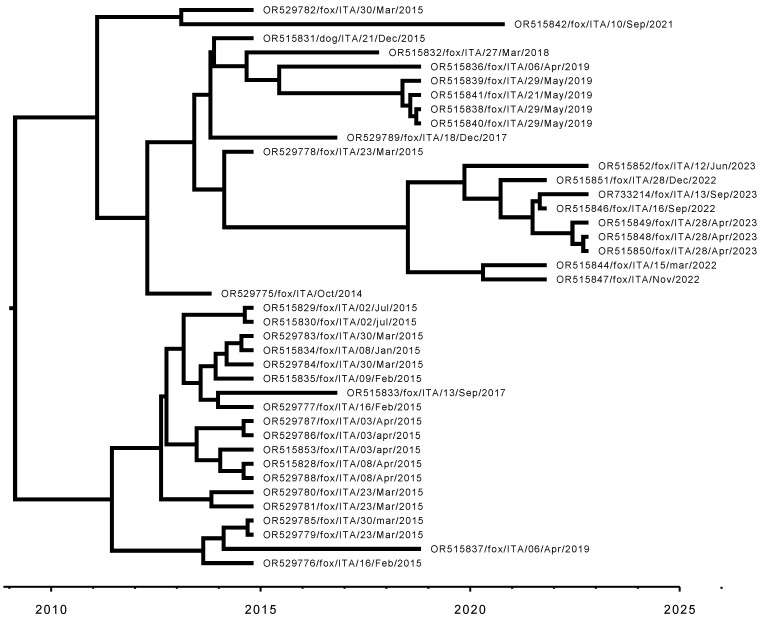
Bayesian reconstruction of the time-calibrated phylogeny for all Sardinian CDV sequences studied in this work, obtained from the BEAST posterior as Maximum Clade Credibility tree with median node heights.

**Figure 9 animals-14-03134-f009:**
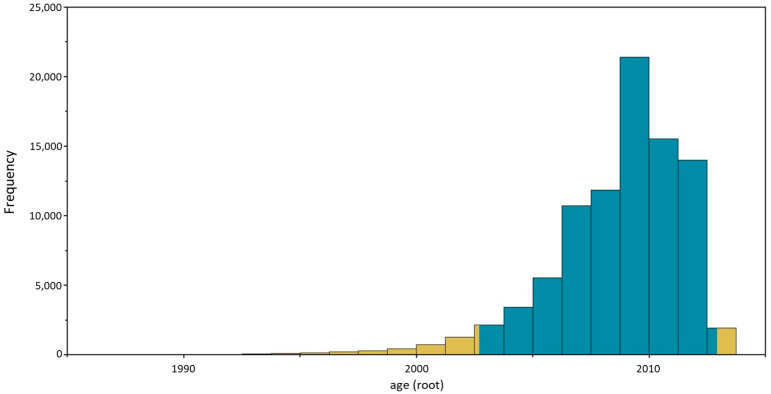
Bayesian reconstruction of the posterior distribution for the date of the Most Recent Common Ancestor of the time-calibrated phylogeny for all Sardinian CDV sequences studied in this work. The blue area represents the 95% Highest Posterior Density (HPD) interval.

**Figure 10 animals-14-03134-f010:**
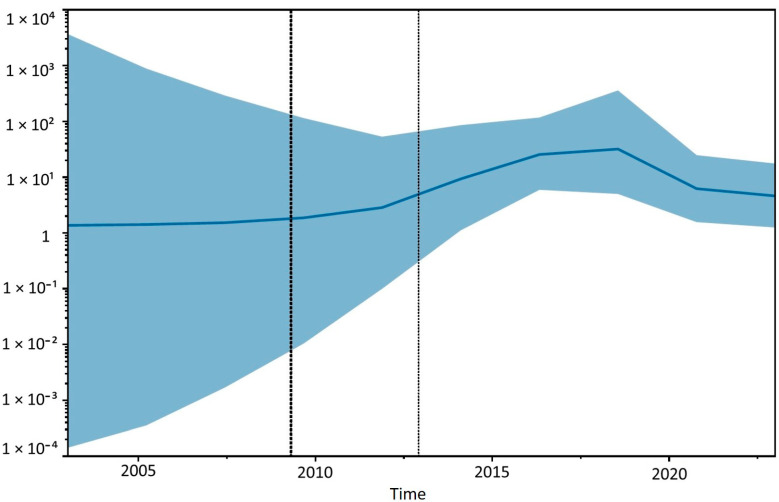
Bayesian skyGrid reconstruction of the time-dependent effective population size of CDV in Sardinia. The continuous line shows the posterior mean log effective population size, while the blue shadow represents the corresponding 95% HPD. The dotted vertical lines represent the 95% HPD for the date of the root of the tree.

**Table 1 animals-14-03134-t001:** Information on hemagglutinin GenBank codes of Sardinian CDV strains detected in the present study.

Strain ID	Sample	Accession Number	Country	Collection Date	Host
Sardinian 1	Brain	OR492629	Italy	3 October 2014	Fox
Sardinian 2	Spleen	OR529775	Italy	3 October 2014	Fox
Sardinian 3	Brain	OR529776	Italy	16 February 2015	Fox
Sardinian 4	Lung	OR529777	Italy	16 February 2015	Fox
Sardinian 5	Lung	OR529778	Italy	23 March 2015	Fox
Sardinian 6	Brain	OR529779	Italy	23 March 2015	Fox
Sardinian 7	Brain	OR529780	Italy	23 March 2015	Fox
Sardinian 8	Lung	OR529781	Italy	23 March 2015	Fox
Sardinian 9	Brain	OR529782	Italy	30 March 2015	Fox
Sardinian 10	Brain	OR529783	Italy	30 March 2015	Fox
Sardinian 11	Lung	OR529784	Italy	30 March 2015	Fox
Sardinian 12	Eyelid	OR529785	Italy	30 March 2015	Fox
Sardinian 13	Brain	OR529786	Italy	3 April 2015	Fox
Sardinian 14	Brain	OR529787	Italy	3 April 2015	Fox
Sardinian 15	Brain	OR529788	Italy	8 April 2015	Fox
Sardinian 16	Lung	OR515828	Italy	8 April 2015	Fox
Sardinian 17	Brain	OR515829	Italy	2 July 2015	Fox
Sardinian 18	Lung	OR515830	Italy	2 July 2015	Fox
Sardinian 19	Brain	OR515831	Italy	21 December 2015	Dog
Sardinian 20	Brain	OR515832	Italy	27 March 2018	Fox
Sardinian 21	Brain	OR515833	Italy	13 September 2017	Fox
Sardinian 22	Brain	OR529789	Italy	18 December 2017	Fox
Sardinian 23	Brain	OR515834	Italy	8 January 2015	Fox
Sardinian 24	Lung	OR515835	Italy	9 February 2015	Fox
Sardinian 25	Brain	OR515836	Italy	6 April 2019	Fox
Sardinian 26	Lung	OR515837	Italy	6 April 2019	Fox
Sardinian 27	Lung	OR515838	Italy	29 May 2019	Fox
Sardinian 28	Brain	OR515839	Italy	29 May 2019	Fox
Sardinian 29	Eyelid	OR515840	Italy	29 May 2019	Fox
Sardinian 30	Lung	OR515841	Italy	21 May 2019	Fox
Sardinian 31	Spleen	OR515842	Italy	10 September 2021	Fox
Sardinian 33	Brain	OR515844	Italy	15 March 2022	Fox
Sardinian 35	Lung	OR515846	Italy	16 September 2022	Fox
Sardinian 36	Brain	OR515847	Italy	28 November 2022	Fox
Sardinian 37	Brain	OR515848	Italy	28 April 2023	Fox
Sardinian 38	Lung	OR515849	Italy	28 April 2023	Fox
Sardinian 39	Eyelid	OR515850	Italy	28 April 2023	Fox
Sardinian 40	Brain	OR515851	Italy	28 December 2022	Fox
Sardinian 41	Brain	OR515852	Italy	12 June 2023	Fox
Sardinian 43	Eyelid	OR515853	Italy	3 April 2015	Fox
Sardinian 44	Brain	OR733214	Italy	13 September 2023	Fox
Sardinian 45	Lung	OR733215	Italy	13 September 2023	Fox

**Table 2 animals-14-03134-t002:** Primer set used to amplify the CDV complete H gene.

Primer Set	Primer Sequence	Size	References	Annealing Temperature (C°)
1st PCR: gF6708	5′ TCTAACCAGATCCTTGAGAC 3′	926 bp	This study	56°
1st PCR: gH7633rev	5′ TGGTGGGAATATGTCACCTCT 3′			
2nd PCR: CDVH 388 for	5′ GAATTCGACTTCCGCGATCTCC 3′	968 bp	This study	50°
2nd PCR: CDVH 1335 rev	5′ ATGGTAAGCCATCCGGAGTTC 3′			
3rd PCR: GH 8360 for	5′ GGTCCGTTATACTGAATGG 3′	792 bp	This study	53°
3rd PCR: gL 9151 rev	5′ AGGAGCTGATAGTTATGTC 3′			

**Table 3 animals-14-03134-t003:** Mean amino acid and nucleotide divergence between sequences from different clades.

**Amino Acid Divergence**			
	Sardinia	Central Italy	Central Europe
Sardinia		4.73%	5.52%
Central Italy			5.34%
Central Europe			
**Nucleotide Divergence**			
	Sardinia	Central Italy	Central Europe
Sardinia		5.03%	
Central Italy			5.20%
Central Europe	5.01%		

## Data Availability

The sequences of the CDV hemagglutinin gene obtained during the present study are available in the GenBank nucleotide sequence database under/with the following accession numbers: OR492629; OR529775; OR529776; OR529777; OR529778; OR529779; OR529780; OR529781; OR529782; OR529783; OR529784; OR529785; OR529786; OR529787; OR529788; OR515828; OR515829; OR515830; OR515831; OR515832; OR515833; OR529789; OR515834; OR515835; OR515836; OR515837; OR515838; OR515839; OR515840; OR515841; OR515842; OR515844; OR515846; OR515847; OR515848; OR515849; OR515850; OR515851; OR515852; OR515853; OR733214; OR733215.

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
