# Peer review of "Canine Distemper Virus in Sardinia, Italy: Detection and Phylogenetic Analysis in Foxesâ€"

_animals, 2024, doi:10.3390/ani14213134_

Round 1
Reviewer 1 Report
Comments and Suggestions for Authors
3214194: Canine Distemper Virus in Sardinia (Italy): detection and Phylogenetic Analysis in foxes and dogs
The authors collected organs from one dog and 41 foxes (n=41). Routine histopathology was done to determine the pattern of organ lesions identified in the collected tissues. IHC was done to detect CDV antigens within the organs valuated. Phylogenetic analyses were done to determine the possible origin of CDV infection. However, the results were poorly presented, particularly the histopathological and immunohistochemical findings, which are fundamental to support the molecular results. Consequently, I cannot recommend this manuscript for publication in its current form. Accordingly, I am providing a list of issues that must be clarified before I can recommend this paper for publication.
Major problems
L27-28. The histopathologic findings are not supportive of infections by CDV (see details below)
L48, a reference is needed for this historical information
L49-50. The disease does not have tropism for tissues; this is an attribute of the virus. Please correct accordingly.
L66-68. A reference is needed to support this information
Table 1. This list of organs collected is not consistent with what was described at L86-88.
2.3. The authors collected tissues from 42 animals. However, there is no mention of how many animals contained CDV nucleic acids at 3.3. additionally, the authors must indicate from which specific organs sequencing and the phylogenetic analyses were done
L114-115. were these samples collected in duplicate?
L150-153. The histopathologic inflammation must be improved. How did you arrive at a diagnosis of "chronic interstitial bronchopneumonia”? i.e., what histological alterations were observed in the lungs?
The lesions described in the CNS are very confused and are not consistent with infections by CDV.
Legend Fig. 2. The authors must provide information as to which inflammatory cells demonstrated positive immunoreactivity within the Brain. How was "high" labelling determined at the cerebellum as compared to other organs. d, inflammatory cells are not being shown in the image provided. The image should be close so that the immunoreactivity can be easily appreciated. It is not clear if the results presented at 3.1. were observed in all animals. This must be made clear since 42 animals were used during this study,
L246-247. I am unable to understand the meaning or implication of this sentence.
L248-249. The histological and IHC findings were poorly presented and cannot be used to correlate the molecular data.
L251-252. This is pure speculation; the authors have not provided any evidence to support this theory
The conclusion is not substantiated by the results, since there is no mention of the number of animals that were effectively infected or died due to CDV-related infections. This can only be achieved by the pathological observations, which were poorly presented.
Minor issues
Introduction; Must be divided into paragraphs for clarity and easy understanding.
L44-45. threatening the extinction of those in danger; this seems a bit over exaggerated.
L87. which histochemical analysis was done during this study?
Author Response
Dear Reviewer 1,
Thank you very much for your revision and advice. Please find below our responses to your suggestions. We used the RED font to evidence the changes we made in the manuscript according to your comments.
Major problems
Question: L27-28. The histopathologic findings are not supportive of infections by CDV (see details below)
Answer: We agree with the referee's concern. We improved the histological and immunohistochemical data. We rewrote the histological descriptions and inserted the results of the performed CD immunophenotyping to characterize the inflammatory cells. Finally, four new picture panels were created to describe our results better and support the CDV etiology.
Question: L48, a reference is needed for this historical information
Answer: Added.
Question: L49-50. The disease does not have tropism for tissues; this is an attribute of the virus. Please correct accordingly.
Answer: Corrected.
Question: L66-68. A reference is needed to support this information
Answer: Added.
Question: Table 1. This list of organs collected is not consistent with what was described at L86-88.
Answer: We specified that not all 4 samples were analyzed for each animal, but that one was chosen based on the best sequencing results (see lines 97-100).
Question: 2.3. The authors collected tissues from 42 animals. However, there is no mention of how many animals contained CDV nucleic acids at 3.3. additionally, the authors must indicate from which specific organs sequencing and the phylogenetic analyses were done
Answer: the samples in Table 1 are those selected for sequencing because they were more informative upon amplification and provided a more punctual sequence. The answer to the previous question also made the concept clearer within the paper.
Question: L114-115. were these samples collected in duplicate?
Answer: No. To clarify the number and type of samples, we added a new sentence to the text on lines 97-100.
Question: L150-153. The histopathologic inflammation must be improved. How did you arrive at a diagnosis of "chronic interstitial bronchopneumonia”? i.e., what histological alterations were observed in the lungs? The lesions described in the CNS are very confused and are not consistent with infections by CDV.
Answer: We improved the histological description by adding more details, and we supported them by testing CDV, CD 3, CD163, and CD79 antibodies in several selected brain and lung sections.
Question: Legend Fig. 2. The authors must provide information as to which inflammatory cells demonstrated positive immunoreactivity within the Brain. How was "high" labelling determined at the cerebellum as compared to other organs. d, inflammatory cells are not being shown in the image provided. The image should be close so that the immunoreactivity can be easily appreciated
It is not clear if the results presented at 3.1. were observed in all animals. This must be made clear since 42 animals were used during this study.
Answer: We deleted this Figure and added 4 new panels with new legends that better describe the severity of lesions and the immunoreactivity patterns.
Of the 42 animals sampled for the work, the brain and lungs of 30 animals with the addition of the lungs of five other foxes were used for histological analysis
Question: L246-247. I am unable to understand the meaning or implication of this sentence.
Answer: necessary clarification added (“with high probability”).
Question: L248-249. The histological and IHC findings were poorly presented and cannot be used to correlate the molecular data.
Answer: We improved the histological and IHC description by adding more details. In this new text, we have highlighted the hallmarks of CDV infections in the affected organs to support our diagnosis.
Question: L251-252. This is pure speculation; the authors have not provided any evidence to support this theory
Answer: We agree with the referee that it is a speculation. We deleted this sentence.
Question: The conclusion is not substantiated by the results, since there is no mention of the number of animals that were effectively infected or died due to CDV-related infections. This can only be achieved by the pathological observations, which were poorly presented.
Answer: Of the 42 animals sampled for the work, the brain and lungs of 30 animals with the addition of the lungs of five other foxes were used for histological analysis. We improved the histological and IHC description by adding more details. In this new text, we have highlighted the hallmarks of CDV infections in the affected organs to support our diagnosis.
Minor issues
Question: Introduction; Must be divided into paragraphs for clarity and easy understanding.
Answer: Done, we divided the introduction into four parts.
Question: L44-45. threatening the extinction of those in danger; this seems a bit over exaggerated.
Answer: Corrected with “that could lead to”.
Question: L87. which histochemical analysis was done during this study?
Answer: We added more details of our IHC investigation, see the paragraph 2.1(lines 129-134/ 143-148).
Reviewer 2 Report
Comments and Suggestions for Authors
In this study, Authors described the canine distemper virus infection in foxes from Sardinia, Italy, through the clinical and anatomo-histopathological finding and their phylogenetic analysis. For this purposes, twenty-one co-Authors collected clinical data, described anatomo-pathological lesions, and obtained and analysed the CDV H gene sequences. Already well note clinical signs and lesions were reported, along with quite divergent H gene sequences, supposing the occurrence of a new sub-lineage. Being, limited and not updated the current data on CDV in wildlife, this study aims to update the literature on CDV infection in foxes and on the molecular features of detected CDV strains. As presented, this study could be more informative and better structured. It could also include more reference literature to support the description. I have below included few general comments and suggestions for the Authors.
Major comments:
- Title: title includes the word “dogs” even if in this study only one CDV H gene sequence from a single dog was included in the analysis. This weakness not only affects the main study (as this only sequence was considered as a potential link between the dog-wildlife interface) but also does not support the proposed title.
- Lines 64-66: this description appears as reductive, compared to the text at lines 230-237, and do not consider the host species. Moreover, reference [37] could be included at line 66.
- Lines 70-72: Are Authors sure that CDV infection does appear to have not a particular clinical impact on foxes?
- Par.2.1: Did Authors collected samples from died foxes and dog, as intuitively deduced from this paragraph, or had they also included in this study analysed sick live animals as reported at lines 155-147? What is reported at the result section is not coherent with the description reported in the M/M section paragraph.
- Par.2.4: Authors included in the analyses a dataset of international sequences, but this dataset and its selection criteria were not included within the manuscript. Are Authors sure to have included all relevant Italian CDV strain sequences?
- Par.3.2: following a previous comment, the H gene sequence of a CDV strain detected in a dog suddenly appears here in the manuscript, without previous detailed information. How could Authors speculate that this finding is suggestive of the virus transmission through the host jump from foxes to dogs? Moreover, data reported in Figure 6 could be considered as referred to the “population size” of mirror the frequency of detection and sequencing? Are these data referred to the results of this study or considered other sampling in the same region in the considered timeframe?
- Discussion, lines 230-244: this part of the manuscript is highly similar to that of another homologous paragraph of a recent paper (Biomolecular Analysis of Canine Distemper Virus Strains in Two Domestic Ferrets (Mustela putorius furo). Vet Sci. 2023 May 26;10(6):375. doi: 10.3390/vetsci10060375), not cited in this manuscript, despite it is one of the few recent papers on CDV strains in wildlife in Europe. I suggest to carefully consider this comment and revise the manuscript accordingly.
- Which criteria were used to propose a new lineage? Should this consider as a divergent lineage or sub-lineage?
Minor comments:
- Line 4: what “#” associated with the first and second Authors’ names does mean?
- Line 22: I suggest to use low-case letters for “Distemper” and “Canine Distemper Virus”.
- Line 24: I suggest to replace “distemper” with “CDV” as it is referred to the virus and not to the disease.
- Line 25: I suggest to replace “the present” with the last year of detection.
- Line 27: “were usually reported” is a generic description or is referred to these cases?
- Line 28: I suggest to replace “histologic” with “histopathological examination”
- Line 30: I suggest to add “the” before “haemagglutinin”
- Line 33: sequences usually are not sampled. Please, check the use of the term “sampled”.
- Line 36: please, consider to replace “new” with “newly proposed”
- Lines 37-38: following a previous major comment, the circulation of this CDV strains in the canine species suddenly appear here in the text, not supported by specific evidence.
- Line 43: Are Authors sure for the use of the “highly prevalent” definition?
- Line 44: I suggest to add a comma after “carnivores”.
- Line 46: I suggest to consider to replace “extinction of those in danger” with “endangered species”
- Line 52: Please, use italics characters for “Morbillivirus”.
- Line 66: despite it is almost clear, I suggest to specify the animal species along with the scientific name.
- Lines 72-47: I suggest to consider to replace “Detecting it early in wild species” with “Early detection in wildlife”; I suggest to remove “through…organs” and “even..lesions” because are not strictly necessary; I also suggest to add “CDV” before “strains”.
- Line 75: I suggest to replace “distemper” with “CDV” since it is referred to the virus and not to the disease.
- Lines 77-78: The text “that…classification” appears as not necessary, since it refers to the premise at lines 58-64 and it is not a viral classification criterium.
- Line 80: “tipify”? “pathological characteristics of the viral strains”?
- Line 84: Usually, it is better to not begin a sentence with a number: I suggest to consider to change with “A total number of 42 animals..”. I also suggest to include more details on the geographical region of origin of these animals.
- Line 85: Did Authors collect eyelids or conjunctival swabs or discharge?
- Line 89: I suggest to add “CDV” before “strains”; please, consider to remove “with …used”, because not necessary, and to replace with “detected”.
- Table 1: I suggest to replace “GB#” with “accession number”; some collection dates are incomplete.
- Line 12: “an ideal..phylogeny” appears as redundant.
- Lines 121-123: Did authors used for downstream analyses the H gene sequences or included the flanking 5’ and 3’ genes fragments? If the first is correct, is this description necessary?
- Table 2: I suggest to replace “PCR steps” with “Primer set”, since these are not PCR steps but primer sets; I suggest to include “size” and “refences”, including in the respective columns the bp size and, if these sequences were newly designed, the text “this study”; finally, I suggest to include the annealing temperature in a separate column.
- Line 134: why Authors included references [31] and [32] referred to the BioEdit software? I suggest to add “softwares” after the squared parenthesis.
- Lines 134-135: I suggest to revise “Maximum likelihood was performed”.
- Figure 2: Scale bars are missing in all images.
- Lines 264 and 265: I suggest to add “region” after “Romagna” and “Abruzzo” respectively. Moreover, I suggest to carefully check the correctness of both region names.
Comments on the Quality of English LanguageFew minor edits of English language could be considered.
Author Response
Dear Reviewer 2,
Thank you very much for your revision and advice. Please find below our responses to your suggestions. We used the blue font to evidence the changes we made in the manuscript according to your comments.
Major comments:
Question: Title: title includes the word “dogs” even if in this study only one CDV H gene sequence from a single dog was included in the analysis. This weakness not only affects the main study (as this only sequence was considered as a potential link between the dog-wildlife interface) but also does not support the proposed title.
Answer: we have removed the word “dog” from the title and better explained what we wanted to say about the connection between the two species.
Question: Lines 64-66: this description appears as reductive, compared to the text at lines 230-237, and do not consider the host species. Moreover, reference [37] could be included at line 66.
Answer: We have included the reference citations and the host species at line 66.
The sentence from the discussion “Within the Europe/South America-1 lineage, two distinct subclades have been identified in northern Italy in recent years: clade a, which originates from ancient epidemic waves in both domestic and wild carnivores, and clade b, which consists of CDV strains most likely imported from the Balkan region.” has been added to the introduction
The discussion has been reworded.
Question: Lines 70-72: Are Authors sure that CDV infection does appear to have not a particular clinical impact on foxes?
Answer: We added explanation and references at lines 81-84.
Question: Par.2.1: Did Authors collected samples from died foxes and dog, as intuitively deduced from this paragraph, or had they also included in this study analysed sick live animals as reported at lines 155-147? What is reported at the result section is not coherent with the description reported in the M/M section paragraph.
Answer: The animals were recovered already dead or died within 72 hours. All animals were therefore subjected to necropsy (see lines149-159: “All foxes died within 2 to 48 hours in the recovery center”).
Question: Par.2.4: Authors included in the analyses a dataset of international sequences, but this dataset and its selection criteria were not included within the manuscript. Are Authors sure to have included all relevant Italian CDV strain sequences?
Answer: We think we have included in the dataset all the Italian H gene sequences intact and correctly deposited on genebank.
Question: Par.3.2: following a previous comment, the H gene sequence of a CDV strain detected in a dog suddenly appears here in the manuscript, without previous detailed information. How could Authors speculate that this finding is suggestive of the virus transmission through the host jump from foxes to dogs?
Answer: The addition of the canine sequence was aimed at understanding possible connections from a phylogenetic point of view with fox sequences. Further sampling of infected dogs in Sardinia and Italy would be needed to corroborate or dispute it. At the moment, according to the authors, our suggestion is supported by the maximum parsimony criterion. Our key finding is the position of the dog sequence in the molecular tree in Figure 7. There are three possible interpretations of this finding:
1) either the host of the MRCA of Sardinia-Wildlife is a dog, implying at least two host jumps from dogs to foxes within Sardinia-Wildlife (and another jump between Sardinia-Wildlife and Italy-Wildlife);
2) or it is a fox, implying at least one jump from foxes to dog within Sardinia-Wildlife;
3) or it is a different species, for example, a badger, implying at least two jumps to dogs/foxes (and another jump within Italy-Wildlife).
Hence, the analysis of host jumps in the subtree formed by Sardinia-Wildlife and the sister clade Italy-Wildlife suggests that the second explanation is the most parsimonious one.
Question: Moreover, data reported in Figure 6 could be considered as referred to the “population size” of mirror the frequency of detection and sequencing? Are these data referred to the results of this study or considered other sampling in the same region in the considered timeframe?
Answer: we do not understand what happened but, compared to the submitted file, the reviewers received the file with a missing figure and with confusing captions. We have now fixed it as originally submitted and think this may clear up the reviewer's justified doubt.
Question: Discussion, lines 230-244: this part of the manuscript is highly similar to that of another homologous paragraph of a recent paper (Biomolecular Analysis of Canine Distemper Virus Strains in Two Domestic Ferrets (Mustela putorius furo). Vet Sci. 2023 May 26;10(6):375. doi: 10.3390/vetsci10060375), not cited in this manuscript, despite it is one of the few recent papers on CDV strains in wildlife in Europe. I suggest to carefully consider this comment and revise the manuscript accordingly.
Answer: This is a pertinent objection. As suggested, the work has been cited, and the period has been revised.
Question: Which criteria were used to propose a new lineage? Should this consider as a divergent lineage or sub-lineage?
Answer: We used the criteria related to nucleotide and amino acid divergence compared to the closest subclade. As emphasized in Figure 7, the divergence between Sardinia-Wildlife and the closest subclade (that we called Italy-Wildlife) or the rest of the Europe-Wildlife sequences, is comparable to the divergence between Asia-6 and Asia-4. Also, note that Sardinia-Wildlife forms a monophyletic clade within the rest of Europe-Wildlife (which would be paraphyletic). Splitting both Sardinia-Wildlife and Italy-Wildlife from Europe-Wildlife would result in all three clades being monophyletic.
Of course, other researchers may debate if would be better classified both Sardinia-Wildlife and Asia-6 as subclades. This is unfortunately a complicated question for all viruses – even for very well-sampled and well-studied viruses such as SARS-CoV-2 – and it would require a proper re-evaluation of the taxonomy of CDV in light of sampling biases and other considerations. However, there is a general consideration that supports our claim: even looking at the full CDV tree in Figure 7, it is clear that the basal branch of Sardinia-Wildlife is one of the longest branches in the whole tree, i.e. Sardinia-Wildlife shows one of the highest levels of divergence from other clades (and the same is true for Italy-Wildlife.) Hence, unless intermediate sequences are found, Sardinia-Wildlife (and Italy-Wildlife) should be classified as a clade under the most reasonable choices of criteria.
Minor comments:
Question: Line 4: what “#” associated with the first and second Authors’ names does mean?
Answer: Done; we apologize for the oversight.
Question: Line 22: I suggest to use low-case letters for “Distemper” and “Canine Distemper Virus”.
Answer: Done.
Question: Line 24: I suggest to replace “distemper” with “CDV” as it is referred to the virus and not to the disease.
Answer: Done.
Question: Line 25: I suggest to replace “the present” with the last year of detection.
Answer: Done.
Question: Line 27: “were usually reported” is a generic description or is referred to these cases?
Answer: It is a generic description.
Question: Line 28: I suggest to replace “histologic” with “histopathological examination”
Answer: Done.
Question: Line 30: I suggest to add “the” before “haemagglutinin”
Answer: Done.
Question: Line 33: sequences usually are not sampled. Please, check the use of the term “sampled”. Answer: Changed with “analysed from”
Question: Line 36: please, consider to replace “new” with “newly proposed”
Answer: Changed.
Question: Lines 37-38: following a previous major comment, the circulation of this CDV strains in the canine species suddenly appear here in the text, not supported by specific evidence.
Answer: In the light of the justified suggestions, received from the reviewers, the presence of the canine sequence is now included in the work in support of the spill-over hypothesis between the domestic and wild species, already ascertained in earlier work now cited, and from the analysis of the amino acid profile of our sequences reported in discussion at lines 281-291.
Question: Line 43: Are Authors sure for the use of the “highly prevalent” definition?
Answer: The definition is cited in Reference 1 (specifically in the abstract and line one of the introduction)
Question: Line 44: I suggest to add a comma after “carnivores”.
Answer: Done.
Question: Line 46: I suggest to consider to replace “extinction of those in danger” with “endangered species”
Answer: Done.
Question: Line 52: Please, use italics characters for “Morbillivirus”.
Answer: Done.
Question: Line 66: despite it is almost clear, I suggest to specify the animal species along with the scientific name.
Answer: Done.
Question: Lines 72-77: I suggest to consider to replace “Detecting it early in wild species” with “Early detection in wildlife”; I suggest to remove “through…organs” and “even..lesions” because are not strictly necessary; I also suggest to add “CDV” before “strains”.
Answer: Done.
Question: Line 75: I suggest to replace “distemper” with “CDV” since it is referred to the virus and not to the disease.
Answer: Done.
Question: Lines 77-78: The text “that…classification” appears as not necessary, since it refers to the premise at lines 58-64 and it is not a viral classification criterium.????
Answer: Deleted.
Question: Line 80: “tipify”? “pathological characteristics of the viral strains”?
Answer: Changed
Question: Line 84: Usually, it is better to not begin a sentence with a number: I suggest to consider to change with “A total number of 42 animals.”
Answer: Done.
Question: I also suggest to include more details on the geographical region of origin of these animals.
Answer: we have added an explanatory map of the areas where sick animals were found.
Question: Line 85: Did Authors collect eyelids or conjunctival swabs or discharge?
Answer: Only eyelids.
Question: Line 89: I suggest to add “CDV” before “strains”; please, consider to remove “with …used”, because not necessary, and to replace with “detected”.
Answer: Done.
Question: Table 1: I suggest to replace “GB#” with “accession number”; some collection dates are incomplete.
Answer: Done.
Question: Line 12: “an ideal..phylogeny” appears as redundant.
Answer: Removed.
Question: Lines 121-123: Did authors used for downstream analyses the H gene sequences or included the flanking 5’ and 3’ genes fragments? If the first is correct, is this description necessary?
Answer: the first hypothesis is correct. The sentence in lines 121-123 has been deleted.
Question: Table 2: I suggest to replace “PCR steps” with “Primer set”, since these are not PCR steps but primer sets; I suggest to include “size” and “refences”, including in the respective columns the bp size and, if these sequences were newly designed, the text “this study”; finally, I suggest to include the annealing temperature in a separate column.
Answer: Done.
Question: Line 134: why Authors included references [31] and [32] referred to the BioEdit software? I suggest to add “softwares” after the squared parenthesis.
Answer: The word “softwares” has been added and the References have been corrected.
Question: Lines 134-135: I suggest to revise “Maximum likelihood was performed”.
Answer: Revised.
Question: Figure 2: Scale bars are missing in all images.
Answer: We added the missing scale bars in all new images.
Question: Lines 264 and 265: I suggest to add “region” after “Romagna” and “Abruzzo” respectively.
Answer: Done.
Question: Moreover, I suggest to carefully check the correctness of both region names.
Answer: Done, the names are correct.
Reviewer 3 Report
Comments and Suggestions for Authors
Dear authors,
I read and reviewed the manuscript "Canine Distemper Virus in Sardinia (Italy): detection and Phylogenetic Analysis in foxes and dogs."
My suggestions are below:
- Lines 22 and 27:
- Remove "the".
- On line 27, add "was" after "necropsy" to ensure correct verb tense: "...was necropsied."
- Lines 38-39:
- Rephrase suggestion: "Given the missing complete genome sequencing, the evidence seems to suggest not a new CDV lineage but rather a subgroup within the existing European lineage, as the authors also note in lines 202-203."
- Line 146:
- Clarification needed: "Given the symptom of hemorrhagic diarrhea, why was PCR not performed for canine parvovirus, considering the possibility of co-infection? Please also specify whether the diarrhea was with digested blood or fresh."
- Lines 150-158:
- Structural reordering: "Please describe the histological and IHC results in order, starting with the findings in the lungs and then proceeding to those in the brain."
- Lines 171-176:
- Technical addition: "In Figure 2, please include a scale bar and specify the magnification in the figure legends, adhering to the journal’s guidelines."
- Lines 204-209, 268-272:
- Conceptual rewrite: Please rewrite these sections considering the hypothesis of a sublineage rather than a new lineage, supported by additional references for comparison. The current data, particularly the absence of complete genome sequencing, do not substantiate the hypothesis of a new CDV lineage but rather suggest a subgroup of the known European lineage. Moreover, the limited number of samples undermines the strength of this hypothesis."
Sincerely
Comments on the Quality of English LanguageThe manuscript would benefit from a thorough review by an English language expert to ensure accuracy and clarity of the manuscript
Author Response
Dear Reviewer 3,
Thank you very much for your revision and advice. Please find below our responses to your suggestions. We used the green font to evidence the changes we made in the manuscript according to your comments.
Question: - Lines 22 and 27:
- Remove "the".
- On line 27, add "was" after "necropsy" to ensure correct verb tense: "...was necropsied."
Answer: Done.
Question: Lines 38-39:
- Rephrase suggestion: "Given the missing complete genome sequencing, the evidence seems to suggest not a new CDV lineage but rather a subgroup within the existing European lineage, as the authors also note in lines 202-203."
Answer: We would like to be able to leave our considerations in the light of the following explanation.
We used the criteria related to nucleotide and amino acid divergence compared to the closest subclade. As emphasized in Figure 7, the divergence between Sardinia-Wildlife and the closest subclade (that we called Italy-Wildlife) or the rest of the Europe-Wildlife sequences is comparable to the divergence between Asia-6 and Asia-4. Also, note that Sardinia-Wildlife forms a monophyletic clade within the rest of Europe-Wildlife (which would be paraphyletic). Splitting both Sardinia-Wildlife and Italy-Wildlife from Europe-Wildlife would result in all three clades being monophyletic. Of course, other researchers may debate whether it would be better to classify both Sardinia-Wildlife and Asia-6 as subclades. This is unfortunately a complicated question for all viruses – even for very well-sampled and well-studied viruses such as SARS-CoV-2 – and it would require a proper re-evaluation of the taxonomy of CDV in light of sampling biases and other considerations. However, there is a general consideration that supports our claim: even looking at the full CDV tree in Figure 7, it is clear that the basal branch of Sardinia-Wildlife is one of the longest branches in the whole tree, i.e. Sardinia-Wildlife shows one of the highest levels of divergence from other clades (and the same is true for Italy-Wildlife). Hence, unless intermediate sequences are found, Sardinia-Wildlife (and Italy-Wildlife) should be classified as a clade under the most reasonable choices of criteria.
Question: Line 146: Clarification needed: "Given the symptom of hemorrhagic diarrhea, why was PCR not performed for canine parvovirus, considering the possibility of co-infection? Please also specify whether the diarrhea was with digested blood or fresh."
Answer: To clarify this point, we changed the word “diarrhea” to ”haematochezia”.
We underlined that samples were submitted to routinary panel of virological and bacteriological examinations. No other positivity was determined in our samples.
We also added the sentence “The foxes that showed gastrointestinal signs were also submitted to virological investigation panel regarding Parvovirus, Coronavirus and bacteriological exames.” at lines 146-148, and the sentence “No other etiological agents were revealed in the samples” at line 236.
Question- Lines 150-158: Structural reordering: "Please describe the histological and IHC results in order, starting with the findings in the lungs and then proceeding to those in the brain." - Lines 171-176: Technical addition: "In Figure 2, please include a scale bar and specify the magnification in the figure legends, adhering to the journal’s guidelines."
Answer: Question on Lines 150-158: we have rewritten and rearranged the results in the text and added 4 new figures.
Question on lines 171-176: we have added the missing scale bars in all new images.
Question: Lines 204-209, 268-272:- Conceptual rewrite: Please rewrite these sections considering the hypothesis of a sublineage rather than a new lineage, supported by additional references for comparison. The current data, particularly the absence of complete genome sequencing, do not substantiate the hypothesis of a new CDV lineage but rather suggest a subgroup of the known European lineage. Moreover, the limited number of samples undermines the strength of this hypothesis."
Answer: We would like to be able to leave our considerations in the light of the following explanation.
We used the criteria related to nucleotide and amino acid divergence compared to the closest subclade. As emphasized in Figure 7, the divergence between Sardinia-Wildlife and the closest subclade (that we called Italy-Wildlife) or the rest of the Europe-Wildlife sequences is comparable to the divergence between Asia-6 and Asia-4. Also, note that Sardinia-Wildlife forms a monophyletic clade within the rest of Europe-Wildlife (which would be paraphyletic). Splitting both Sardinia-Wildlife and Italy-Wildlife from Europe-Wildlife would result in all three clades being monophyletic. Of course, other researchers may debate whether it would be better to classify both Sardinia-Wildlife and Asia-6 as subclades. This is unfortunately a complicated question for all viruses – even for very well-sampled and well-studied viruses such as SARS-CoV-2 – and it would require a proper re-evaluation of the taxonomy of CDV in light of sampling biases and other considerations. However, there is a general consideration that supports our claim: even looking at the full CDV tree in Figure 7, it is clear that the basal branch of Sardinia-Wildlife is one of the longest branches in the whole tree, i.e. Sardinia-Wildlife shows one of the highest levels of divergence from other clades (and the same is true for Italy-Wildlife). Hence, unless intermediate sequences are found, Sardinia-Wildlife (and Italy-Wildlife) should be classified as a clade under the most reasonable choices of criteria.
Round 2
Reviewer 1 Report
Comments and Suggestions for Authors
Canine Distemper Virus in Sardinia (Italy): detection and Phylogenetic Analysis in foxes
Although the authors have attended to most of my recommendations and the quality of the paper has been greatly improved, there are still a few that must be rectified before I can recommend this paper for publication.
Major problems
L39-40 Alter to read: Necropsy and histopathological examination revealed interstitial pneumonia and mild, diffuse non-purulent meningoencephalitis with multifocal, moderate to severe demyelination, respectively.
L148, alter virological to molecular investigation.
L149-150. The authors must provide more details of the assays done so that readers can understand how these analyses were done to detect additional pathogens. This is of fundamental importance to either include or exclude the participation of another infectious disease pathogen.
L214, remove "Broncho" to read interstitial pneumonia
L 218-2919 alter to read "blood vessels was observed in some pulmonary sections".
L259, remove broncho to read interstitial pneumonia
L268, How can you confirm this information when there is no mention of this investigation (by IHC or molecular testing) in the MM section. Since the information provided in the revised manuscript lacks specific details (see previous comment)
Minor problems
Paper title: Remove the brackets from Italy and include a comma to read .. in Sardinia, Italy
Keywords: Maybe substitute CDV (already given in the title) for canine morbillivirus and include other words that are not within the title of the submitted paper.
L96, alter to read " ...Sardinia, Italy ..."
L201. .alter were determined to were observed
Author Response
Dear Reviewer 1,
Thank you very much for your revision and advice. Please find below our responses to your suggestions. We used the RED font to evidence the changes we made in the manuscript according to your comments.
Major problems
Question: L39-40 Alter to read: Necropsy and histopathological examination revealed interstitial pneumonia and mild, diffuse non-purulent meningoencephalitis with multifocal, moderate to severe demyelination, respectively.
Answer: Done.
Question: L148, alter virological to molecular investigation.
Answer: Done.
Question: L149-150. The authors must provide more details of the assays done so that readers can understand how these analyses were done to detect additional pathogens. This is of fundamental importance to either include or exclude the participation of another infectious disease pathogen.
Answer: We have added more details with references, in the MM section.
Question: L214, remove "Broncho" to read interstitial pneumonia.
Answer: Done.
Question:L 218-219 alter to read "blood vessels was observed in some pulmonary sections".
Answer: Done.
Question: L259, remove broncho to read interstitial pneumonia
Answer: Done.
Question: L268, How can you confirm this information when there is no mention of this investigation (by IHC or molecular testing) in the MM section. Since the information provided in the revised manuscript lacks specific details (see previous comment)
Answer: We have added details about the investigations in the MM section.
Minor problems
Question: Paper title: Remove the brackets from Italy and include a comma to read .. in Sardinia, Italy
Answer: Done.
Question: Keywords: Maybe substitute CDV (already given in the title) for canine morbillivirus and include other words that are not within the title of the submitted paper.
Answer: We have substituted all keywords as suggested
Question: L96, alter to read " ...Sardinia, Italy ..."
Answer: Done.
Question: L201. .alter were determined to were observed
Answer: Changed.
Reviewer 2 Report
Comments and Suggestions for Authors
With this version of the manuscript, Authors have reviewed almost all observed criticisms, substantially improving the overall description. I have below included a few additional general comments and suggestions for the Authors.
Major comments:
- Line 98: It is not clear why these references have been included in this sentence, referring to the sequence analysis performed in this study.
- Line 177: Please, check these references: one is not referred to the MEGA 7.0 version and other two are referred to the same reference.
- Lines 333-336: these two references seem almost identical, and the meaning appears redundant. They could be merged into a unique sentence.
- Reference list: please, check the newly added references, since retained a different format [43], have not correctly listed [1. and 1.] or are double [20 and 1.].
Minor comments:
- Line 29: I suggest to replace “our” with “the”
- Lines 31-32: I suggest to remove “to limit its spread and”, since this cannot be achieved by the “continue monitoring”.
- Line 39: I suggest to remove “usually” and “in the foxes” since this information is referred to the tested animals.
- Lines 44-45: As reported, “The genetic…dog one” seems not referred to the genomic sequences of detected CDV strains.
- Keywords: I suggest to use a lower case letter for “Distemper”.
- Line 59: I suggest to replace “the” with “some”.
- Lines 61-62: This sentence seems more linked to the following ones and, therefore, I suggest to move the sentence to the next paragraph.
- Line 100: “better” could be moved before “characterize”.
- Line 107: I suggest to move the comma before “and” just after this word.
- Figure 1: Is this map only referring to the sampling area of the (41) foxes?
- Line 117: I suggest to add “of” after “codes”.
- Lines 150: canine parvovirus and canine coronavirus?
- Line 480: please, check “corpse will be handed”
Comments on the Quality of English LanguageI suggest a further minor English language editing
Author Response
Dear Reviewer 2,
Thank you very much for your comments and advice. Please find below our responses to your suggestions. We used the blue font to evidence the changes we made in the manuscript according to your comments.
Major comments:
Question: Line 98: It is not clear why these references have been included in this sentence, referring to the sequence analysis performed in this study.
Answer: We have moved the corrected citations between the sentence at line 96 and the previous one.
Question: Line 177: Please, check these references: one is not referred to the MEGA 7.0 version and other two are referred to the same reference.
Answer: We have corrected in the text and the reference list by deleting the reference [36] (related to MEGA 5.0) and the [38] (the double related to MEGA 7.0.)
Question: Lines 333-336: these two references seem almost identical, and the meaning appears redundant. They could be merged into a unique sentence.
Answer: We have corrected and merged the two references into one.
Question: Reference list: please, check the newly added references, since retained a different format [43], have not correctly listed [1. and 1.] or are double [20 and 1.].
Answer: We have checked and corrected the formatting.
Minor comments:
Question: Line 29: I suggest to replace “our” with “the”
Answer: Replaced.
Question: Lines 31-32: I suggest to remove “to limit its spread and”, since this cannot be achieved by the “continue monitoring”.
Answer: Removed.
Question: Line 39: I suggest to remove “usually” and “in the foxes” since this information is referred to the tested animals.
Answer: Both were removed.
Question: Lines 44-45: As reported, “The genetic…dog one” seems not referred to the genomic sequences of detected CDV strains.
Answer: We removed “and the dog one”
Question: Keywords: I suggest to use a lower case letter for “Distemper”.
Answer: We have removed “Distemper” and replace with “Morbillivirus”.
Question: Line 59: I suggest to replace “the” with “some”.
Answer: Replaced.
Question: Lines 61-62: This sentence seems more linked to the following ones and, therefore, I suggest to move the sentence to the next paragraph.
Answer: We have moved the sentence.
Question: Line 100: “better” could be moved before “characterize”.
Answer: Done.
Question: Line 107: I suggest to move the comma before “and” just after this word.
Answer: Done.
Question: Figure 1: Is this map only referring to the sampling area of the (41) foxes?
Answer: Yes. We have changed the opening sentence in the figure caption.
Question: Line 117: I suggest to add “of” after “codes”.
Answer: Added.
Question: Lines 150: canine parvovirus and canine coronavirus?
Answer: Yes we have added more details with references.
Question: Line 480: please, check “corpse will be handed”
Answer: We have checked and corrected the sentence.
Minor editing of English language required:
corrections made to lines:
35
56-57
59-60
69
103-105
110
145-146
194-195
203-204
339
371
379